

# Mott transition and pseudogap of the square-lattice Hubbard model: Results from center-focused cellular dynamical mean-field theory

Michael Meixner[1], Henri Menke[1], Marcel Klett[1], Sarah Heinzelmann[2], Sabine Andergassen[3], Philipp Hansmann[4] and Thomas Schäfer[1⋆]

**1** Max Planck Institute for Solid State Research, Stuttgart, Germany
**2** Institute for Theoretical Physics and Center for Quantum Science,
University of Tübingen, Tübingen, Germany
**3** Institute for Solid State Physics and Institute of Information Systems Engineering,
Vienna University of Technology, Vienna, Austria
**4** Department of Physics, Friedrich-Alexander-University Erlangen-Nürnberg,
Erlangen, Germany

⋆ t.schaefer@fkf.mpg.de

## Abstract

The recently proposed center-focused post-processing procedure [Phys. Rev. Res. 2, 033476 (2020)] of cellular dynamical mean-field theory suggests that central sites of large impurity clusters are closer to the exact solution of the Hubbard model than the edge sites. In this paper, we systematically investigate results in the spirit of this center-focused scheme for several cluster sizes up to $8 \times 8$ in and out of particle-hole symmetry. First we analyze the metal-insulator crossovers and transitions of the half-filled Hubbard model on a simple square lattice. We find that the critical interaction of the crossover is reduced with increasing cluster sizes and the critical temperature abruptly drops for the $4 \times 4$ cluster. Second, for this cluster size, we apply the center-focused scheme to a system with more realistic tight-binding parameters, investigating its pseudogap regime as a function of temperature and doping, where we find doping dependent metal-insulator crossovers, Lifshitz transitions and a strongly renormalized Fermi-liquid regime. Additionally to diagnosing the real space origin of the suppressed antinodal spectral weight in the pseudogap regime, we can infer hints towards underlying charge ordering tendencies.

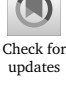

# 1 Introduction

The reason why a material changes its electronic properties from a metallic to an insulating state, and the way how this transition exactly happens, is a fundamental research question in condensed matter physics. Apart from rooting in the underlying crystal structure (described by the usual band picture), metal to insulator transitions (MIT) can be driven by the mutual interactions of electrons and quantum many-body fluctuations [1], a hallmark of strongly correlated systems (in which the common band picture breaks down). Famous examples of such fluctuation-driven transitions and crossovers are the Mott transition in Cr-doped $V_2O_3$ [2, 3] and the pseudogap regime found in layered copper-oxide compounds, the so-called cuprates [4, 5].

The arguably simplest realization of correlation-driven metal-insulator transitions and crossovers in a theoretical model can be found in the Hubbard model [6–9]. Here the MIT

is driven by a shift in the competition of kinetic (electrons hopping on a lattice) and potential energy (included as an effective, purely local approximation of the Coulomb interaction). Despite its simplicity, this model cannot be exactly solved in the experimentally relevant dimensions of two and three. However, the last decades have seen an enormous improvement of our understanding of this model, both analytically [10] and computationally [11].

Specifically, for the understanding of metal-insulator transitions in strongly correlated systems, a big step forward was the conception of the dynamical mean-field theory (DMFT) [12–14], which is able to connect the non-interacting (itinerant) with the strong coupling (atomic) limit in a single non-perturbative framework. DMFT fully takes temporal correlations into account, however, neglects spatial ones. The latter become particularly important in the case of low dimensionality (or coordination number) at low temperatures and, in terms of physical phenomena, for the description of, e.g., Fermi arcs and, hence, the pseudogap. These non-local correlations can be included into the DMFT framework by cluster [15] and diagrammatic [16] extensions of DMFT. Within the cluster extensions an interacting cluster of $N_c$ sites is embedded into a non-interacting, self-consistently determined bath. Spatial correlations are then treated exactly up to the cluster size, which serves as a control parameter, i.e., in the limit $N_c \to \infty$ the solution of the cluster is identical to the solution of the original lattice problem in the thermodynamic limit. Technically, the cluster can be established in real space within the cellular DMFT (CDMFT) [17–19]), or, alternatively, the self-energy can be patched in momentum space in the dynamical cluster approximation (DCA) [15]).

In practical terms the size of the cluster is limited due to computational constraints, which usually depend on the regime of the phase diagram to be investigated (temperature, doping, interaction parameters). This makes it particularly hard to extrapolate data obtained from these finite-size clusters to the thermodynamic limit. Specifically for the real space CDMFT, many important results for the two-dimensional Hubbard model have been obtained using a $N_c = 2 \times 2$ plaquette cluster:[1] analyses of the Mott transition [25–28] and the pseudogap [29, 30], as well as its interplay with superconductivity [31–35], and antiferromagnetism [35–37].

Recent computational improvements of the quantum impurity solvers allowed for larger cluster sizes than the plaquette. This not only extends the range of correlations which are progressively included in the calculation, but also allows for the distinction of center and edge (see also sketches of clusters in App. E). Klett, et al. [38] demonstrated that the extrapolation to the thermodynamic limit converges faster with $N_c$ if the outer edge of the respective cluster is neglected in the post-processing, a procedure coined 'center-focused extrapolation'. These results suggest, that quantities obtained from the center of the cluster in general give better approximations to the exact result of the thermodynamic limit. For this reason, averages over the entire cluster (e.g., when periodizing to obtain momentum-dependent quantities) should be avoided.

In this paper we systematically study the two-dimensional Hubbard model within the CDMFT on intermediate- to large-sized clusters, and apply the center-focused post-processing. We investigate the metal-insulator transition for the particle-hole symmetric case, before we study the appearance of a pseudogap, the Fermi-surface reconstruction and charge fluctuations for the doped Hubbard model with a finite $t'$ within this technique. The manuscript is structured as follows: In Section 2 we introduce the model and its parameters and detail the algorithm used for our center-focused CDMFT. In Section 3 we analyze the phase diagram of the half-filled Hubbard model on a square lattice with $t' = 0$, also using large clusters. We analyze the consequences of doping in Section 4 and a non-zero $t'$ in Section 5, respectively. In the latter we also comment on possible indicators of charge fluctuations. Eventually, in Section 6 we draw conclusions and give an outlook on further applications of the algorithm.

---

[1]With the notable exception of the investigation of different periodization schemes in [20]. For systematic DCA studies on these topics see for instance [15, 21–24].

## 2 Model and methods

### 2.1 The two-dimensional Hubbard model

We study the two-dimensional Hubbard model on the square-lattice

$$H = -t \sum_{\langle i,j \rangle, \sigma} c_{i,\sigma}^\dagger c_{j,\sigma} - t' \sum_{\langle\langle i,j \rangle\rangle, \sigma} c_{i,\sigma}^\dagger c_{j,\sigma} + U \sum_i n_{i,\uparrow} n_{i,\downarrow} - \mu \sum_{i,\sigma} n_{i,\sigma}, \qquad (1)$$

where $c_{i,\sigma}$ ($c_{i,\sigma}^\dagger$) represents the site-dependent fermionic creation (annihilation) operator of an electron with spin $\sigma \in \{\uparrow, \downarrow\}$ on site $i$, $t$ is the nearest-neighbour (nn) hopping, $t'$ the second-nearest-neighbour (2nn) hopping, $U$ the local Coulomb repulsion and $\mu$ the chemical potential. We set $t = 1$ throughout the paper and give all energies in this unit. Apart from Section 5, where we analyze a non-zero $t' = -0.25t$, $t' = 0$ is considered throughout this paper.

### 2.2 The cellular dynamical mean-field theory

We analyze this model by applying the cellular dynamical mean-field theory (CDMFT), a real space cluster extension [15] of the dynamical mean-field theory (DMFT) [12–14]). In CDMFT the auxiliary impurity Anderson model does not correspond to a single lattice site but to a real space super-cell containing $N_c$ unit-cells (in our case containing a single site). As in single-site DMFT, the auxiliary problem is found by converging a self-consistent loop around the condition

$$G_{i,j}^{\text{loc}}(i\omega_n) := \sum_{\vec{k} \in \text{RBZ}} \left[ (i\omega_n - \mu)\delta_{i,j} - \varepsilon_{i,j}(\vec{k}) - \Sigma_{i,j}^{\text{imp}}(i\omega_n) \right]^{-1} \overset{!}{=} G_{i,j}^{\text{imp}}(i\omega_n), \qquad (2)$$

where $i, j$ are cluster-site indices, $\vec{k}$ is the wave-vector in the reduced Brillouin zone (RBZ) of the super-lattice, $\varepsilon_{i,j}(\vec{k})$ the super-cell dispersion relation and $\Sigma_{i,j}^{\text{imp}}(i\omega_n)$ the site- and Matsubara frequency dependent self-energy obtained from the impurity cluster embedded in a bath. Via this cluster self-energy, the local Green function of the CDMFT contains non-local correlations up to the length of the real space cluster. Hence, the CDMFT is a controlled technique which recovers the exact result of the lattice problem in the limit $N_c \to \infty$. The fulfillment of the self-consistency condition (2) is achieved by an iterative procedure (starting with an initial guess of the self-energy), during which the cluster Weiss-field is computed by a matrix-valued Dyson equation:

$$\left[ \mathcal{G}^0(i\omega_n) \right]_{i,j}^{-1} = \left[ G^{\text{loc}}(i\omega_n) \right]_{i,j}^{-1} + \Sigma^{\text{imp}}(i\omega_n)_{i,j}. \qquad (3)$$

Together with the value of the Hubbard interaction $U$ the Weiss field defines the impurity model, which in this study is solved by means of continuous-time quantum Monte Carlo in its interaction expansion (CT-INT) [39,40]). We employ the solver provided as an application in TRIQS [41]. We stress that the choice of CT-INT as an impurity solver was crucial[2] to reach the cluster sizes discussed in the present manuscript.

### 2.3 Periodization and center-focused post-processing

As a real space technique involving a system of finite size, CDMFT breaks the translation invariance of the lattice problem in its thermodynamic limit. The translation invariance can be restored by a so-called periodization procedure *after* convergence of the CDMFT self-consistency

---

[2]CT-INT QMC does not display a sign problem in the particle-hole symmetric case, significantly speeding up calculations. The parameter set with the highest computational cost is the data-point at $T = 0.05t$, $U = 5t$, $t' = -0.25t$, $n = 0.85$ where the DMFT cycle started from the converged result of n=0.875 and consumed approximately 300'000 core hours throughout 20 DMFT loops until convergence was reached while a perturbation order of 1300 was necessary to yield an acceptable sign.

cycle [20]. In essence, this periodization corresponds to a Fourier transformation, where there is a certain arbitrariness in the choice of which quantity $Q$ is transformed from real to momentum space:

$$
\begin{aligned}
Q(\vec{k}, i\omega_n) &= \sum_{i,j} f_{[i,j]}(\vec{k}) Q_{i,j}(i\omega_n) \\
&= \sum_{i,j} e^{i\vec{r}_{i,j}\vec{k}} Q_{i,j}(i\omega_n),
\end{aligned}
\tag{4}
$$

where $f$ represents the $\vec{k}$-momentum dependent Fourier coefficient between the cluster sites $i$ and $j$ and $Q_{i,j}(i\omega_n)$, the respective quantity to be periodized on the Matsubara axis. In [38,42] it has been shown that the self-energy obtained from a center-focused extrapolation converges faster with the cluster size than the periodization schemes previously introduced in the literature. Hence, for all single-particle quantities in momentum space shown in this manuscript, we choose to periodize the center-focused self-energy as a post-processing step:

$$
\begin{aligned}
\Sigma(\vec{k}, i\omega_n) &= \sum_{j} f_{[c,j]}(\vec{k}) \Sigma_{c,j}(i\omega_n) \\
&= \sum_{j} e^{i\vec{r}_{c,j}\vec{k}} \Sigma_{c,j}(i\omega_n),
\end{aligned}
\tag{5}
$$

where $c$ denotes the center site of the cluster. From Eq. (5) The lattice Green function is then computed via

$$
G(\vec{k}, i\omega_n) = \frac{1}{i\omega_n - \mu + \varepsilon_{\vec{k}} - \Sigma(\vec{k}, i\omega_n)},
\tag{6}
$$

from which a proxy of the spectral weight at the Fermi level can be extracted by

$$
A(\vec{k}_F) = -\frac{1}{\pi} \operatorname{Im} G(\vec{k}_F, i\omega_n \to 0),
\tag{7}
$$

for vectors on the Fermi surface

$$
\vec{k}_F \in \mathrm{FS} := \left\{ \vec{k} \in \mathrm{BZ} \mid 0 \overset{!}{=} \tilde{\varepsilon}_k \right\},
\tag{8}
$$

where BZ is the Brillouin zone and $\tilde{\varepsilon}_k = -\mu + \varepsilon_{\vec{k}} - \operatorname{Re}\Sigma(\vec{k}, i\omega_n \to 0^+)$ the renormalized dispersion. We further define the non-interacting dispersion $\tilde{\varepsilon}_{\vec{k}, U=0} = -\mu + \varepsilon_{\vec{k}}$. The extrapolations to $i\omega_n \to 0^+$ are executed as a second order polynomial fit of the first three Matsubara frequencies. As two distinguished points on the Fermi surface we define the nodal point $k_N$, where $k_x^{\mathrm{FS}} = k_y^{\mathrm{FS}}$ and the anti-nodal point, where one direction of $(k_x, k_y) = \vec{k}^{\mathrm{FS}}$ equals either 0 or $\pi$. Throughout the work we define $T^*$ as the one-particle onset temperature of a pseudogap, i.e., the suppression of spectral weight at the anti-node when decreasing the temperature of the system. To amend our analysis, we sometimes show spectral functions on the real axis. These are obtained from the imaginary time Green function data by the maximum entropy (MaxEnt) analytic continuation method [43], available as application[3] of TRIQS [44] with a cost function optimized for the single band case [45]. We restrict ourselves to the paramagnetic solution of CDMFT, except for the determination of antiferromagnetic transition temperatures. For the latter we applied a staggered magnetic field, calculated the magnetic susceptibility in linear response $\chi_m(\mathbf{q} = (\pi, \pi), i\Omega_n = 0)$ and linearly extrapolated its inverse $\chi_m^{-1} \to 0$ to determine $T_{\mathrm{N\acute{e}el}}$.

---

[3]https://triqs.github.io/maxent/latest/guide/tau_maxent.html.

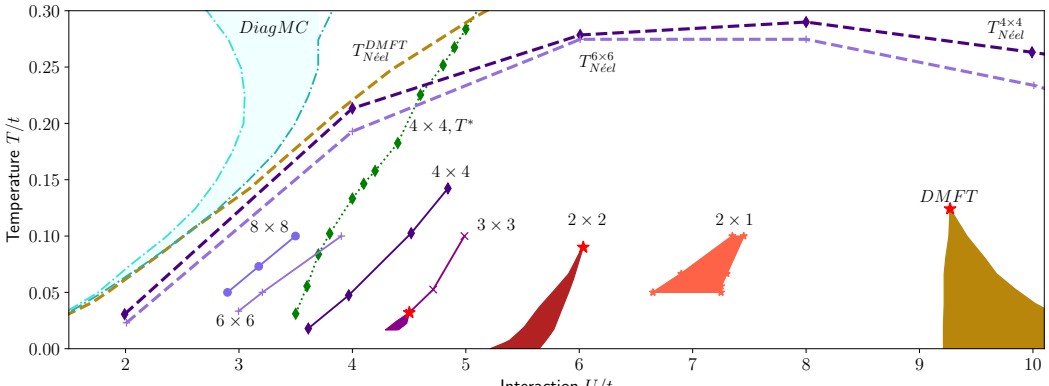

Figure 1: Phase diagram of the two-dimensional Hubbard model on a square lattice for different CDMFT cluster sizes $N_c$. The filled areas indicate metal-insulator coexistence regions, the solid lines indicate Mott crossover lines separating metallic from insulating regimes. Red stars indicate $2^{\text{nd}}$ order critical endpoints of Mott transitions. The dotted green line indicates the opening of the pseudogap for $N_c = 4 \times 4$ at $T < T^*$. The turquoise area shows the metal to non-metal crossover obtained from DiagMC [46,47] for comparison. The dashed lines give the Néel temperatures for the respective cluster sizes. From other sources: DMFT coexistence region [48,49], $2 \times 2$ coexistence region [27], and $T_{\text{Néel}}^{\text{DMFT}}$ [47,50].

# 3 Interaction-driven metal to insulator crossover: Half filling and $t' = 0$

## 3.1 Phase diagram

We start our analysis with $t' = 0$ and half filling, where we first investigate the phase diagram as a function of interaction value $U$, non-zero temperature $T$ and cluster size $N_c$. We restrict ourselves to the (metastable) paramagnetic solution of CDMFT. Fig. 1 shows the respective interaction-driven metal-insulator crossovers and transitions. As already pointed out in previous studies [48,51] single-site DMFT (equivalent to $N_c = 1$ CDMFT) exhibits a first order metal-insulator transition with a second-order critical endpoint at $U_c^{1\times1} \approx 9.3t$ and $T_c^{1\times1} \approx 0.12t$, marked by a red star. At temperatures $T < T_c$ a coexistence region (indicated in brown) with a hysteresis behavior is found. When increasing the cluster size to a dimer $N_c = 2$ and to a $2 \times 2$ plaquette with $N_c = 4$, the interaction values of the respective critical endpoints are reduced considerably, while the critical temperature $T_c$ remains almost the same. The qualitative change in slope of the spinodals of the coexistence region w.r.t. DMFT can be understood by the fact that the increased cluster size allows for a singlet formation including an effective non-zero exchange coupling [52,53].

In contrast to the $2 \times 2$ plaquette, where only the critical interaction $U_c$ is significantly reduced w.r.t. DMFT, for larger cluster sizes also the critical temperature drops abruptly. Above $T_c$ only a metal to insulator crossover (solid line) is found, determined by the center site's spectral weight inflection point (Widom line).

For cluster sizes $N_c \geq 16$, quite remarkably, we could not find a coexistence region (and, hence, a phase transition), but only a metal-insulator crossover line for the accessible temperatures ($T \geq 0.0166t$). We attribute this behavior to more and more extended antiferromagnetic fluctuations, which are progressively included with larger cluster sizes [54]. Although the CDMFT solution is restricted to the paramagnetic metastable case, on the impurity cluster

these fluctuations are fully taken into account. The importance of these fluctuations is immediately apparent when calculating the actual antiferromagnetically ordered solution and determining the Néel temperature for different cluster sizes (dashed lines in Fig. 1). In the limit of $N_c \rightarrow \infty$ the Mermin-Wagner theorem [55] is recovered and ordering at non-zero temperatures is inhibited. However, strong non-local correlations would also persist in the thermodynamic limit [46, 47, 50]. There, these long-ranged antiferromagnetic spin fluctuations drive a metal-insulator crossover, as demonstrated earlier by dynamical vertex approximation (DΓA) [50, 56] and numerically exact diagrammatic Monte Carlo (DiagMC) calculations [46, 47], as well as in analytical work [47, 57–59]. Further, a recent study has pointed out the competition of non-local antiferromagnetic correlations and local Mott physics in a momentum dependent metal-insulator crossover in extensions of DMFT via the D-TRILEX scheme [60].

### 3.2 Determination of the metal-insulator crossover and self-energy analysis

Fig. 2 a) demonstrates the qualitative change in the spectral weight for $T = 0.05t$ in more detail: The $N_c = 2 \times 2$ case exhibits a slow decay in spectral weight when increasing the value of the interaction. At $U \approx 5.8t$ the spectral weight discontinuously drops to almost zero, indicating a manifest phase transition for this cluster size and temperature. In contrast, for the $N_c = 4 \times 4$ size cluster we find that the decrease of the spectral weight is different for non-equivalent sites of the cluster: the central sites become insulating at lower interaction values than the corner and (not shown) edge sites. This can be understood by the fact that the central site has a different coordination number to either (non-interacting) bath sites or (interacting) cluster sites than the corner and edge sites. Closer inspection reveals that the decrease of the corner site spectral weight exhibits three different regimes: one regime, which resembles the shape of the decrease at the central site, an intermediate plateau, and finally a rapid drop. In the spirit of center-focused CDMFT we assign the onset $U_c$ of the metal-insulator crossover shown in Fig. 1 at the inflection point of the central site's spectral weight [32, 61]. Increasing $N_c$ even further monotonously decreases $U_c$, with qualitatively the same shape of the drop of the spectral weight at the center site.

Additional insight into the origin of the metal-insulator crossover and the evolution of the spectral weight with $U/t$ can be gained by an analysis of the cluster self-energies. In Fig. 2 b)–d) we show the real parts (without their respective Hartree contributions) and the imaginary parts of the self-energies as a function of Matsubara indices for three different values of $U/t$ at $T = 0.05t$. For $U = 3.8t$ we observe that the on-site self-energies of both center and corner site show a (renormalized) metallic behavior, i.e., they can be Taylor expanded for small Matsubara frequencies (dashed lines). The nn and 2nn contributions are rather small in both cases, which results in a similar behavior of the spectral weight for both sites. When increasing the interaction to $U = 5t$ a dichotomy between these two sites becomes apparent: while the on-site self-energy of the central site exhibits a divergent behavior for small frequencies, the corner site self-energy is still metallic. Furthermore, when examining the intersite self-energies we see that the 2nn component of the inner $2 \times 2$ plaquette is almost three times larger than the 2nn component of the corner site. Eventually, at $U = 6t$, also the on-site corner self-energy diverges, giving rise to an insulating behavior also there.

Finally, we mention that for $N_c = 4 \times 4$, a true pseudogap can be observed: Upon cooling the system, a full suppression of the antinodal spectral weight concomitant with an increasing nodal spectral weight is found (see Fig. 1). This is in contrast to the $N_c = 2 \times 2$ plaquette, where at half-filling only a momentum differentiation occurs (see Appendix B.1 for the plaquette cluster data). The details of the pseudogap and its doping dependence will be discussed in the following Sections 4 and 5.

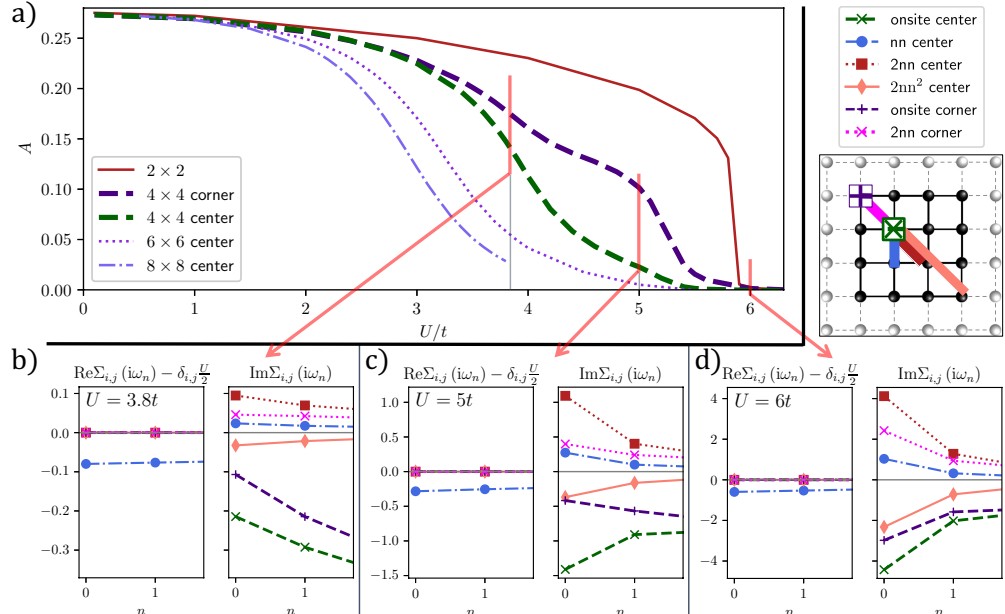

Figure 2: a) Spectral weight as a function of the interaction strength $U/t$ for $T = 0.05t$ and several cluster sizes and inequivalent cluster sites. b)–d) Real and imaginary parts of the cluster self-energies of the $N_c = 4 \times 4$ cluster for $U = 3.8t$ in b), $U = 5t$ in c), and $U = 6t$ in d).

# 4 Doping-driven metal to insulator crossover: $t' = 0$

We now turn to the analysis of single-particle properties upon electron and hole doping. Here we consider results only from the $N_c = 4 \times 4$ cluster at a temperature of $T = 0.05t$. At this temperature, the half-filled system is insulating for $U \geq 4.2t$ in the center-focused CDMFT (see Section 3.2). For our calculations we choose a slightly higher interaction value of $U = 5t$. In this section we consider $t' = 0$, the effects of a finite $t'$ will be addressed in Section 5.

## 4.1 Real space spectral weights and self-energies

Fig. 3 a) shows the spectral weights upon doping resolved for center, edge and corner sites. By construction the observable is symmetric in electron and hole doping at $t' = 0$, so that it is sufficient to discuss only the hole-doped side. The qualitative behavior of the spectral weight agrees for all sites: upon doping, first the spectral weight increases (which relates to the fact that the half-filled case is the most insulating one), before it drops again due to the decreased filling of the system. We can furthermore see a differentiation of the spectral weight of different sites, which is strongest in the half-filled case: the center site is more insulating than the corner site (with the edge site in between). This differentiation is mitigated by increased doping: first, at $n \approx 0.9$ the spectral weights of corner and edge sites become equivalent, then, at $n \approx 0.85$ they merge with the spectral weight of the center site. Fig. 3 b)–c) shows an analogous analysis of the cluster self-energies as for the half-filled case. Three points are noteworthy: (i) Already at $n = 0.95$ the longer-ranged self-energy components have a rather small magnitude compared to the on-site ones. (ii) For this case, the real part of the self-energies have a considerable non-zero contribution, which contrasts the half-filled case. (iii) For large dopings the largest component of the self-energy is the on-site one with absence of site differentiation. We will see later that this corresponds to a conventional Fermi-liquid regime.

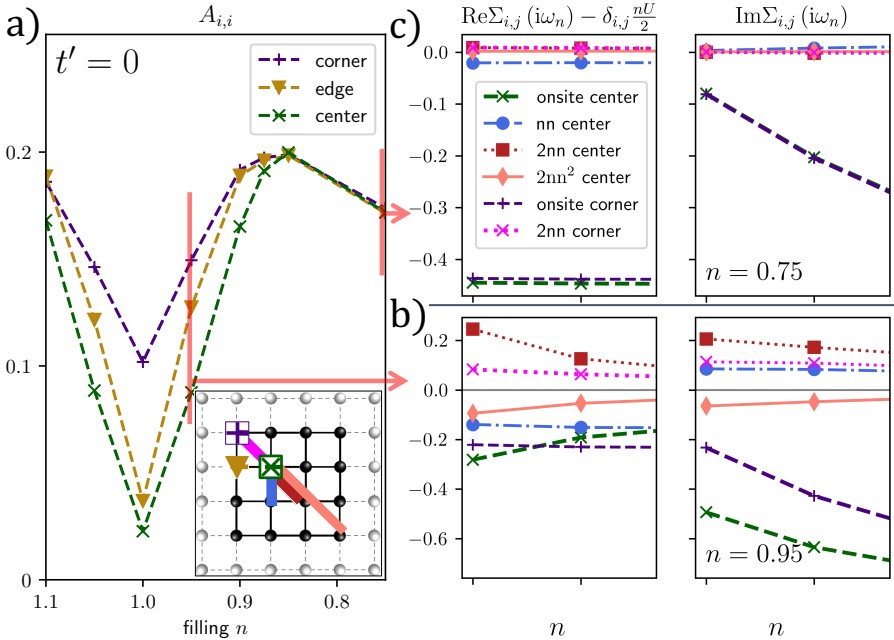

Figure 3: a) Spectral weight as a function of electron and hole doping for $U = 5t$, $T = 0.05t$ and inequivalent cluster sites. b)–c) Analogous self-energy analysis to Fig. 2 b)–d) for dopings $n = 0.95$ in b) and $n = 0.75$ in c).

## 4.2 Momentum space analysis: Fermi surface renormalization, pseudogap, and Fermi liquid

After our real space analysis we now continue with observables in momentum space. To this end we employ a periodization by post-processing the center-focused CDMFT results (see Section 2). Fig. 4 a) shows the evolution of the Fermi surface and the spectral weight (color coded) as a function of the electron filling for $U = 5t$ and $T = 0.05t$. The interacting (dotted-dashed black lines) and non-interacting ($U = 0$) Fermi surfaces (white lines) have been determined as roots of the quasiparticle equation Eq. (8).

Upon hole doping the Fermi surface of the non-interacting system immediately becomes electron-like, i.e., the dispersion $\tilde{\varepsilon}_{(\pi,0),U=0} > 0$, at $(\pi,0)$, resulting in a convex shape of the Fermi surface. Here the seminal Luttinger theorem [62] (strictly valid only at $T = 0$) is broken, indicating the presence of a Luttinger surface [63–65].

In contrast, for the interacting system the Fermi surface becomes hole-like with $\tilde{\varepsilon}_{(\pi,0)} < 0$ for the renormalized dispersion, resulting in a concave shape of the Fermi surface for small dopings. It then undergoes a Lifshitz transition to electron-like for larger doping levels $n \approx 0.85$. These changes of topology of the Fermi surface are in agreement with the findings of [66].

We can infer the degree of metallicity from the temperature dependence of the spectral weight at $\omega = 0$ at the node and antinode, shown in Fig. 4 b) for different dopings. At half filling $n = 1$ we observe a depletion of the spectral weight at the antinode concomitant to an increase at the node upon cooling, which is a single-particle signature of a pseudogap.[4] This notion is confirmed by the analytically continued data for the spectral function at $T = 0.05t$ shown in Fig. 4 c): while one observes a gap at the antinode, a sharp quasiparticle peak remains at the node. The suppression of the spectral weight at the antinode is driven by a pole in its

---

[4]Indeed, a folding of the Fermi-surface at the antiferromagnetic zone boundary is hinted for the half-filled case in Fig. 4 a), allowing an alternative definition of the antinode as the maximum of the spectral weight at the border of the Brillouin zone. From this, a pseudogap can also be inferred, see App. C.

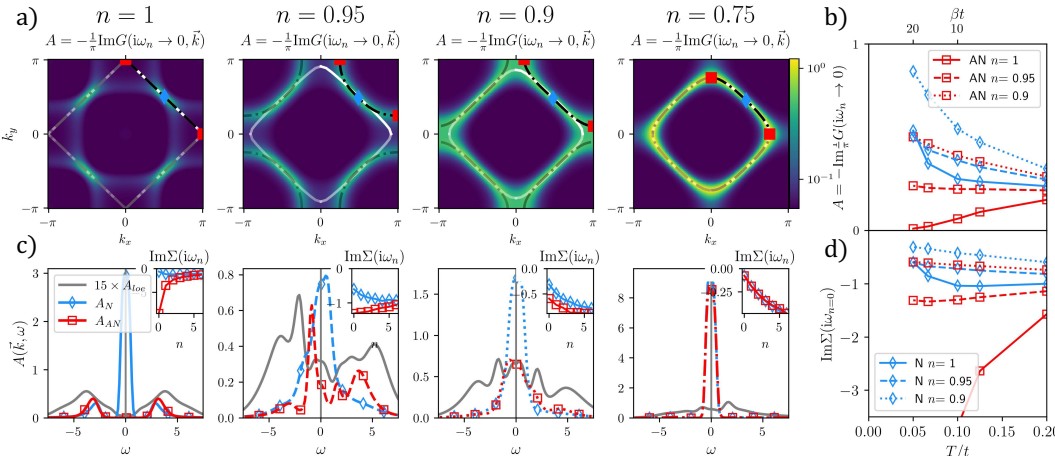

Figure 4: a) Fermi surfaces and spectral weights (color coded), extrapolated from the Matsubara axis for $U = 5t$, $t' = 0$, and $T = 0.05t$ obtained from the $N_c = 4 \times 4$ cluster and center-focused periodization, as a function of the filling $n$. White solid lines indicate the non-interacting Fermi surfaces, black dotted-dashed lines the respective interacting ones. b) The spectral weight at the antinode and node as a function of the temperature and filling. c) Analytically continued [44] spectral function for the antinode (red), node (blue), and the local spectral function (grey). The insets show the imaginary part of the self-energies as a function of the Matsubara index for the antinode and node. d) Imaginary part of the self-energy at the first Matsubara frequency as a function of temperature. Red squares denote the antinode, blue diamonds the node throughout the figure.

self-energy, which originates from accumulations of the real space self-energy contributions (see Section 4.3), as indicated by the data on the Matsubara axis shown in the inset. For $n = 0.95$ the suppression of the spectral weight is not as strong anymore at the antinode, suggesting that the pseudogap closes with increasing hole doping. This is confirmed by the emergent quasiparticle peak at the antinode at $n = 0.9$ and its increasing spectral weight upon cooling. At large hole doping $n = 0.75$, additionally to its now electron-like nature and the absence of a pseudogap, the momentum differentiation on the Fermi surface vanishes.

Let us finally comment here on the nature of the metallic states. As already pointed out, at half filling the metallicity is destroyed at the antinode by a large scattering rate, represented by the imaginary part of the self-energy on the Matsubara axis. Fig. 4 d) shows $\operatorname{Im}\Sigma(T, i\omega_0)$, i.e., for the first Matsubara frequency and as a function of temperature. For a regular Fermi liquid, this quantity should display a linear temperature dependence for all points on the Fermi surface [67]. At $n = 1.0$ the value at the antinode indeed diverges for low $T$, whereas the value at the node is (non-linearly) reduced at low $T$. Interestingly, for $n = 0.95$ $\operatorname{Im}\Sigma(T, i\omega_0)$ is linear at the node, but exhibits non-linear features at the antinode, indicating a momentum-differentiated Fermi liquid at this doping level. At fillings $n \leq 0.9$, $\operatorname{Im}\Sigma(T, i\omega_0)$ is linear at the node and exhibits an offset at $T = 0$ and a constant temperature derivative at the antinode and therefore represents a metal with Fermi-liquid properties at the node. While the qualitative phase diagram agrees with calculations using Diagrammatic Monte Carlo [68], we find the pseudogap crossover and the Lifshitz transitions closer to half filling, i.e., at slightly smaller values of the hole doping.

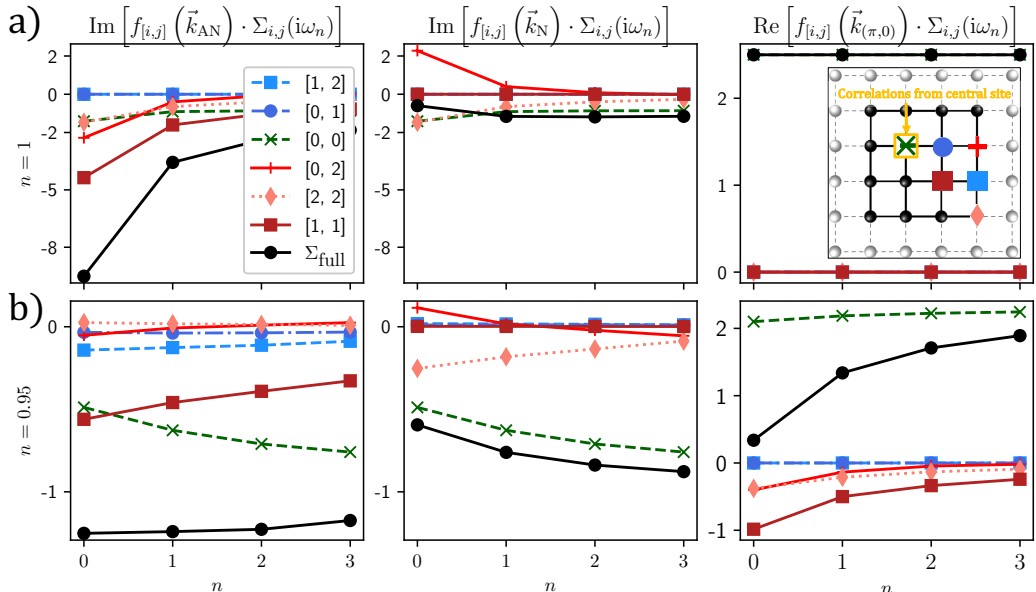

Figure 5: Decomposition of the full momentum-space self-energy $\Sigma_{\text{full}}(\vec{k}, i\omega_n)$ (black line) into contributions from different distances on the lattice, in the center-focused scheme, analyzed for a) filling $n = 1$ and b) filling $n = 0.95$. One line corresponds to the total of all contributions to the self-energy from an intersite-distance of the cluster with the respective form factors $f_{[i,j]}(\vec{k})$. The contributions are analyzed for the antinodal (left), nodal (central), and $(\pi, 0)$ momentum point (right column), respectively.

## 4.3 Real space analysis of momentum space correlations

In order to analyze the origin the of pseudogap and the Fermi surface reconstruction, in this subsection we decompose the self-energy for certain $\vec{k}$-points into the contributions from the center-focused periodization Eq. (5). In Fig. 5 we plot the contributions to the imaginary part of the periodized self-energy. We decompose the full self-energy (black line) into the correlations between the center and other sites of the cluster. We then classify all real space self-energies in this decomposition by the distances $d$ between a respective site and the center site. For instance, the red solid line with a plus marker labelled $[0, 2]$ corresponds to the summed contributions from $\Sigma_{[0,2]}, \Sigma_{[2,0]}, \Sigma_{[-2,0]}, \Sigma_{[0,-2]}$ with their respective Fourier form factors $f_{[i,j]}(\vec{k})$ for a given $\vec{k}$. This allows us to analyze the total contributions from different cluster distances to the real and imaginary parts of the full self-energy $\Sigma_{\text{full}}(\vec{k}, i\omega_n)$ at momentum $\vec{k}$. In the following discussion we discriminate between different distances on the cluster by dividing them into two classes, highlighted by color coding. The first class being combinations of lattice vectors $\vec{R} = (1, 1), (1, -1)$ ("reddish" distances) and the second class of all others ("blueish" distances). This divides our non-local real space correlations into a bipartite checkerboard, shown in the inset of Fig. 5. This divides our non-local real space correlations into a bipartite checkerboard, shown in the inset of Fig. 5. We exclude the center's on-site, i.e. $\vec{R} = (0, 0)$ (green line with marker x), contribution from this colour coding.

We start our analysis with the particle-hole symmetric case with $n = 1$ in a). Here, all self-energies apart from the on-site Hartree contributions are purely imaginary. Investigating the self-energy at the antinode $\vec{k}_{AN} = (0, \pi)$ (left column), we see that only the on-site correlation and the reddish distances contribute to the imaginary part of the self-energy. Interestingly, the latter exceed the on-site correlation considerably, resulting in a strong quasiparticle scattering

in anti-nodal direction, which we attribute, due to the occurring spatial pattern, to enhanced antiferromagnetic fluctuations. At the node $\vec{k}_N = (\pi/2, \pi/2)$ (central column), also exclusively the on-site and the reddish distances contribute. However, compared to the anti-node, the contributions of $[0, 2]$-vectors pick up the opposite sign from the form factors, reducing the nodal scattering rate significantly. This "protection" of the nodal quasiparticles concomitant with the large scattering rate at the antinode eventually results in the formation of a pseudogap. The real part of the self-energy at $\vec{k} = (\pi, 0)$ (right column) only has the on-site Hartree contribution, which corresponds exactly to the chemical potential $\mu = U/2$ in the particle-hole symmetric case. Hence, no contribution to the renormalized dispersion $\tilde{\varepsilon}_k$ occurs, and the anti-node coincides with $\vec{k} = (\pi, 0)$.

We now turn to the analysis of the hole-doped case with filling $n = 0.95$, shown in row b) of Fig. 5. This case corresponds to a reconstructed Fermi surface, where the pseudogap is on the brink of closing. The imaginary part of the self-energy at the anti-node $\vec{k}_{AN} \approx (0.83, \pi)$ only shows contributions from 2nn (dark red) and on-site real space self-energies, while all other distances hardly play any role. This leads to a flat self-energy $\Sigma_{full}(\vec{k}_{AN}, i\omega_n)$ of a strongly renormalized metal at the anti-node. At the nodal point, $\vec{k}_N \approx (1.5155, 1.5155)$, the reddish distances give only small contributions to the imaginary part of the self-energy, in essence given by the on-site contribution. At $\vec{k} = (0, \pi)$ the only contribution comes from the real part of the cluster self-energies. Here, all reddish distances give a strong, negative contribution, drastically reducing the full self-energy at this point. This shifts away the Fermi surface determined by a vanishing of the renormalized dispersion, which is the real space cause of the Fermi surface reconstruction demonstrated in Fig. 4.

## 5 Doping-driven metal to insulator crossover: $t' = -0.25t$

In order to form a closer connection with experiments, we now introduce a second-nearest neighbor (2nn) hopping $t' = -0.25t$ in our square lattice, a value in the ballpark of, e.g., certain cuprate and infinite-layer nickelate quasi two-dimensional compounds [69–72]. Again we analyse real space spectral weights and self-energies before using the center-focused periodization for investigating quantities in momentum space. Regarding the interaction strength we stay again with the computationally feasible value of $U = 5t$, which, as shown below, also reveals a pseudogap upon hole doping (see below).

### 5.1 Real space spectral weights and self-energies

Fig. 6 a) shows the respective spectral weights of center, edge and corner sites for different degrees of doping. As already the non-interacting density of states is particle-hole asymmetric due to the introduction of a non-zero $t'$, also electron and hole doping have different effects. For 10% electron doping, the spectral weight of all sites is equal. When reducing the electron doping below $n = 1.05$, the spectral weight decreases, reaching a minimum at $n = 0.95$ before slowly increasing again, which is different from the $t' = 0$ case, where this minimum was found at half filling. While the qualitative behavior of the spectral weights agree for all sites, again the center site shows the most insulating properties compared to edge and corner sites. For $n \leq 0.9$, the center site is more insulating than the edge and corner sites, of which the latter are of equal spectral weight. No differentiation between sites is visible anymore at $n = 0.75$.

Figs. 6 b)–e) show the corresponding self-energy analysis, as analogously shown for $t' = 0$ in Fig. 3 b)–c). While for $n = 1.05$ only the on-site self-energy contributes, we find a considerable contribution of the real part self-energies of the inner cluster at $n = 1$. This real part contribution decreases when doping towards $n = 0.95$ and is significant only at the zeroth Matsubara frequency for the on-site and 2nn self-energy of the inner cluster. Concomitantly,

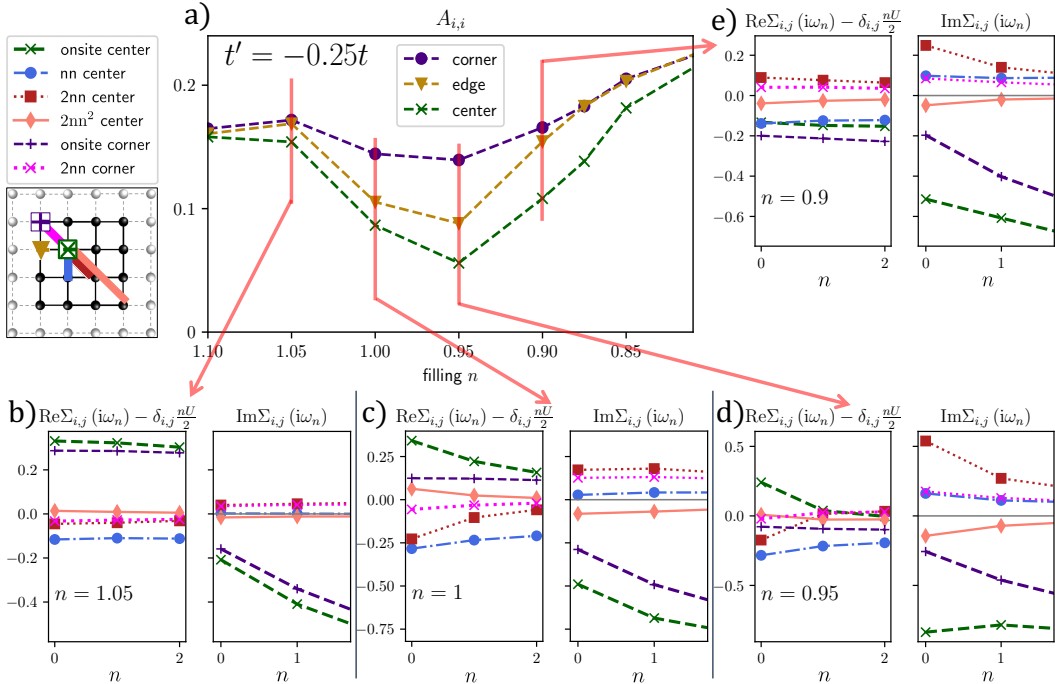

Figure 6: a) Spectral weight as a function of electron and hole doping for $T = 0.05t$ and inequivalent cluster sites. b)–e) real space self-energy analysis for different doping levels.

the imaginary part of the self-energy of the inner cluster qualitatively changes from metallic to insulating. When reducing the filling to $n = 0.9$ the imaginary part of the self-energy reduces slightly, while the real parts do not show much variation with the Matsubara index at small frequencies. This contrasts the $t' = 0$ case where the self-energy showed an insulating imaginary part only at half filling, while a finite real part occurred for slight electron and hole doping.

## 5.2 Momentum space analysis: Fermi surface renormalization, pseudogap, and Fermi liquid

In analogy to Section 4.2 we employ the center-focused post-processing scheme to periodize from real to momentum space for the case $t' = -0.25t$. Fig. 7 a) shows the evolution of the Fermi surface and the spectral weight (color coded) as a function of the electron filling for the more interesting hole doped case. Again, the interacting (dotted-dashed black lines) and non-interacting ($U = 0$) Fermi surfaces (white lines) have been determined by the quasiparticle Eq. (8). At half filling $n = 1$ the Fermi surface of the non-interacting system is hole-like. In contrast, interestingly, the interacting Fermi surface is reconstructed towards electron-like with a reduced volume, again indicating the presence of a Luttinger surface [63–65]. On the almost circular Fermi surface, the spectral weight is distributed uniformly along the Fermi surface. When reducing the filling, the interacting Fermi surface becomes hole-like and the Fermi volume increases. For $n = 0.9$ the spectral weight is strongest at the nodal point (blue diamond marker) and weakest at the anti-node (red square marker). When further hole doping towards $n = 0.85$, this nodal anti-nodal dichotomy reduces, while the Fermi surface topology shows no qualitative change. At strongly reduced filling of $n = 0.75$, the interacting Fermi surface is on the verge of changing back to electron-like again and almost resembles the shape of the non-interacting one. The spectral weight is again distributed uniformly along the Fermi

SciPost Phys. **16**, 059 (2024)

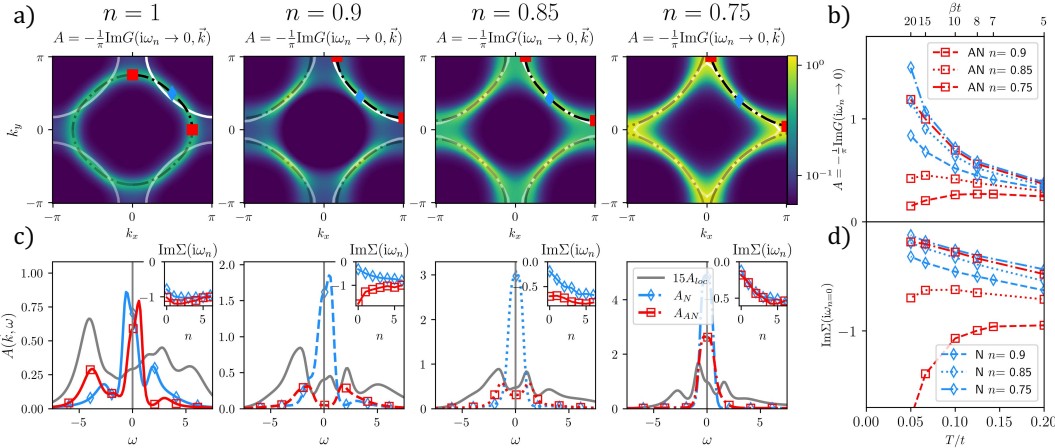

Figure 7: a) Fermi surfaces and spectral weights (color coded), extrapolated from the Matsubara axis for $U = 5t$, $t' = -0.25t$, and $T = 0.05t$ obtained from the $N_c = 4 \times 4$ cluster and center-focused periodization, as a function of the filling $n$. White solid lines indicate the non-interacting Fermi surfaces, black dotted-dashed lines the respective interacting ones. b) The spectral weight at antinode and node as a function of the temperature and filling. c) Analytically continued [44] spectral function for the antinode (red), node (blue), and the local spectral function (grey). The insets show the imaginary part of the self-energies as a function of the Matsubara index for the antinode and node. d) Imaginary part of the self-energy at the first Matsubara frequency as a function of temperature. Red squares denote the antinode, blue diamonds the node throughout the figure.

surface, hinting towards a common Fermi liquid.

The temperature dependence of the spectral weights in Fig. 7 b) demonstrates a reduction of the spectral weight for the anti-node in contrast to an increase for the node at both $n = 0.9$ and $n = 0.85$. This indicates a pseudogap for these fillings. For lower fillings, both nodal- and anti-nodal spectral function increase when decreasing temperature, demonstrating metallic behaviour.

These electronic properties become even more clearly visible in the spectral data obtained from the analytic continuation, see Fig. 7 c). For the half-filled case $n = 1$ we find equally large quasiparticle peaks for node and anti-node, where only the Hubbard-bands are slightly asymmetric and of different value. The self-energies for these points are indeed very similar to that of a strongly renormalized metal. Let us point out here that most likely this regime of the CDMFT phase diagram would be magnetically ordered, if we did not restrict ourselves to the paramagnetic solution of the self-consistency.

For $n = 0.9$ the nodal spectral function exhibits a strong quasiparticle peak and no Hubbard bands, while the anti-nodal spectral function exhibits Hubbard bands and a complete gap at the Fermi level $\omega = 0$. This is supported by the self-energies in the inset: The nodal self-energy is that of a weakly renormalized metal, while the anti-nodal self-energy is clearly divergent, underpinning the conclusion of a pseudogap behavior at this doping level. At reduced filling of $n = 0.85$, we find that the gap in the spectral function for the anti-nodal point is closing and indeed the corresponding self-energy is turning from insulating to metallic. For lower filling a renormalized metal at the anti-node combined with a Fermi-liquid behaviour at the node can be found (not shown). Eventually, at $n = 0.75$ also this momentum differentiation ceases upon doping for the spectral function and the self-energy at the Fermi surface. The first Matsubara frequency criterion in Fig. 7 d) indeed confirms the Fermi-liquid nature of this metallic regime.

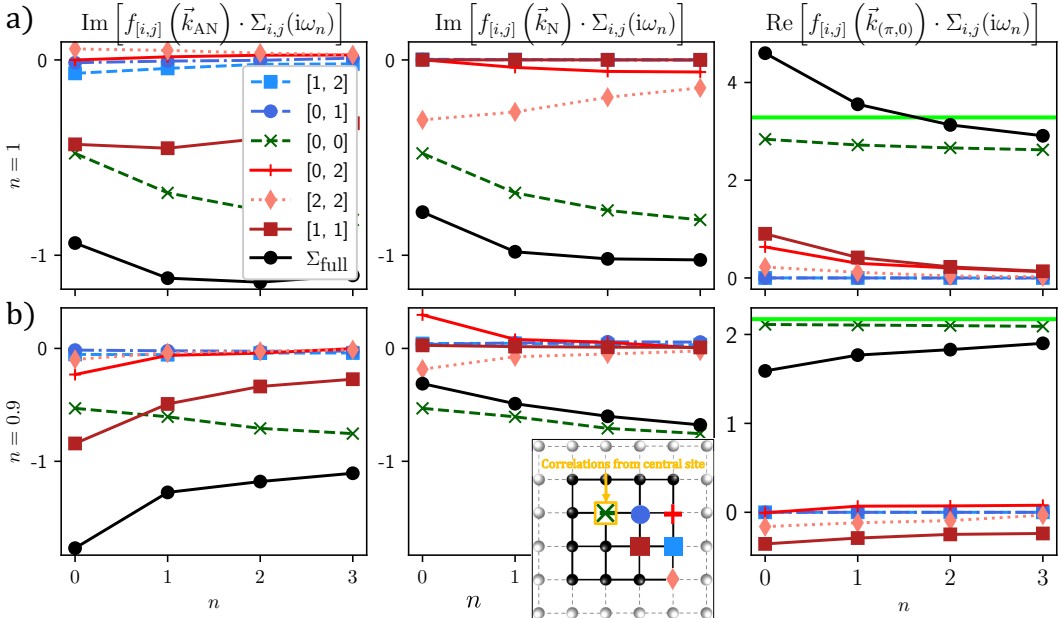

Figure 8: real space analysis of the self-energy contributions at filling a) $n = 1$ and b) $n = 0.9$ for $\beta = 20/t$ and $U = 5t$. The green solid line in the third column marks the bare dispersion plus the chemical potential $\varepsilon_k + \mu$ for the respective case.

## 5.3 Real space self-energy analysis of the Fermi surface renormalization and the pseudogap

In analogy to Section 4.3, we turn to the analysis of real space contributions to the momentum space self-energies in the center-focused post-processing scheme for $t' = -0.25t$ in Fig. 8. Focusing on the exemplary case of the renormalized Fermi surface at $n = 1$ one finds that the imaginary part of the self-energy is almost exclusively composed of the on-site and 2nn self-energies, while for the imaginary part of the nodal self-energy, only the 2nn correlation and the on-site self-energy play a role. For both $\vec{k}$-points, the qualitative shape of the full self-energy is given by their respective on-site contributions. The third column provides the real part of the self-energy at $\vec{k} = (\pi, 0)$, responsible for the renormalization of the Fermi surface. When the full self-energy $\Sigma_{\text{full}}$ exceeds $\varepsilon_k + \mu$ (green bold line), the Fermi surface is electron-like. Here, we find an almost constant on-site contribution, which resembles the Hartree diagram $Un/2$. The full self-energy then has strong contributions from the reddish distances, particularly the 2nn and $[0, 2]$-distance, leading to the change in topology of the Fermi surface. The blueish distances do not play a role due to lattice symmetry $\vec{k} = (\pi, 0)$. Similar to the opening of the pseudogap this change in topology has been attributed to antiferromagnetic fluctuations, shown, e.g., by an effective gauge theory [73–75]. In addition, these spin-density wave ordering tendencies have been shown to be responsible for a charge carrier drop in the two-dimensional Hubbard model [76].

For the more hole doped pseudogap case $n = 0.9$, we find that while the on-site self-energy resembles the one of a renormalized metal, the anti-nodal scattering rate is gapped only by correlations from non-local reddish distances. This contrasts the nodal scattering rate which is reduced by correlations from reddish distances, where particularly the imaginary part of the $[0, 2]$-self-energy leads to a lower scattering rate at the node, and hence a longer lifetime of quasi particles. This relates to recent findings of dual fermions with a boson exchange parquet solver, where spin fluctuations were found to lead to an increased quasiparticle lifetime at the nodal point [77]. The real parts of the non-local self-energies at $\vec{k} = (\pi, 0)$ have now

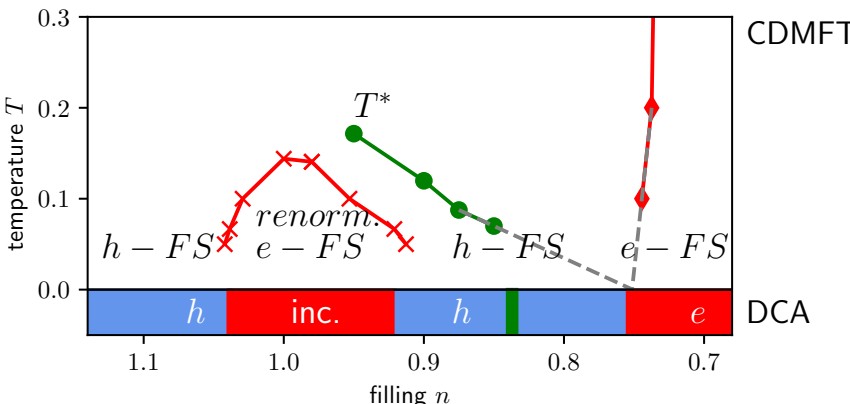

Figure 9: Phase diagram of the two-dimensional Hubbard model for $t' = -0.25t$ at $U = 5t$, obtained from paramagnetically restricted center-focused CDMFT with $N_c = 4 \times 4$. The red crosses indicate a Fermi surface reconstruction, the green circles give dopings and temperatures $T^*$ for which a pseudogap was found for $T < T^*$. Red diamonds give the Lifshitz transition due to decreased filling. The colored bar at the bottom of the figures shows the zero-temperature phase diagram obtained by DCA with $N_c = 8$ at $U = 7t$ (taken from [78]), where the respective Fermi surface topology is indicated by $h$ (hole-like) or $e$ (electron-like), and the pseudogap closing by a vertical bar. The red area around half-filling marks an incoherent (inc.) system, where [78] makes no statement about the Fermi surface topology.

changed signs and turned negative, reducing the real part of the full self-energy below the renormalized dispersion. This leads to a change from an electron like Fermi surface at n=1 to a hole-like Fermi surface at n=0.9. Again, for the renormalization only the reddish distances play a significant role.

## 5.4 Phase diagram and comparison to the dynamical cluster approximation

So far we have restricted our analysis mostly to the doping dependence at a fixed temperature of $T = 0.05t$. We now turn to the temperature dependence of our results and summarize them in the $T$ vs. filling $n$ phase diagram reported in Fig. 9. The red lines indicate the changes of topology of the Fermi surface determined by the sign of the renormalized dispersion at the $\vec{k} = (\pi, 0)$-point, see also Eq. (8). The Fermi surface shows the following features:

(i) A Lifshitz transition can be found (red diamond markers) between a hole-like Fermi surface $n \geq 0.75$ and an electron-like one for lower fillings. The filling for this change in topology varies only slightly with temperature.

(ii) At around half filling $n = 1$,[5] another change in topology is found for $T \leq 0.18t$, where the Fermi surface is reconstructed from hole-like to electron-like (red crosses). Interestingly, in this regime, the Pomeranchuk effect in the double occupancy is absent, see Appendix A.

(iii) For the hole-like Fermi surface in the hole-doped case, a pseudogap opening can be inferred (green circular markers) by the analysis of the temperature dependence of the

---

[5]At slight electron doping $n = 1.02$, we did not observe a pseudogap down to $T = 0.066t$ at $U = 5t$ which contrasts studies for higher interaction values of $U = 7t$ [78] or for the triangular lattice at half filling [24] and for the anisotropic case [79].

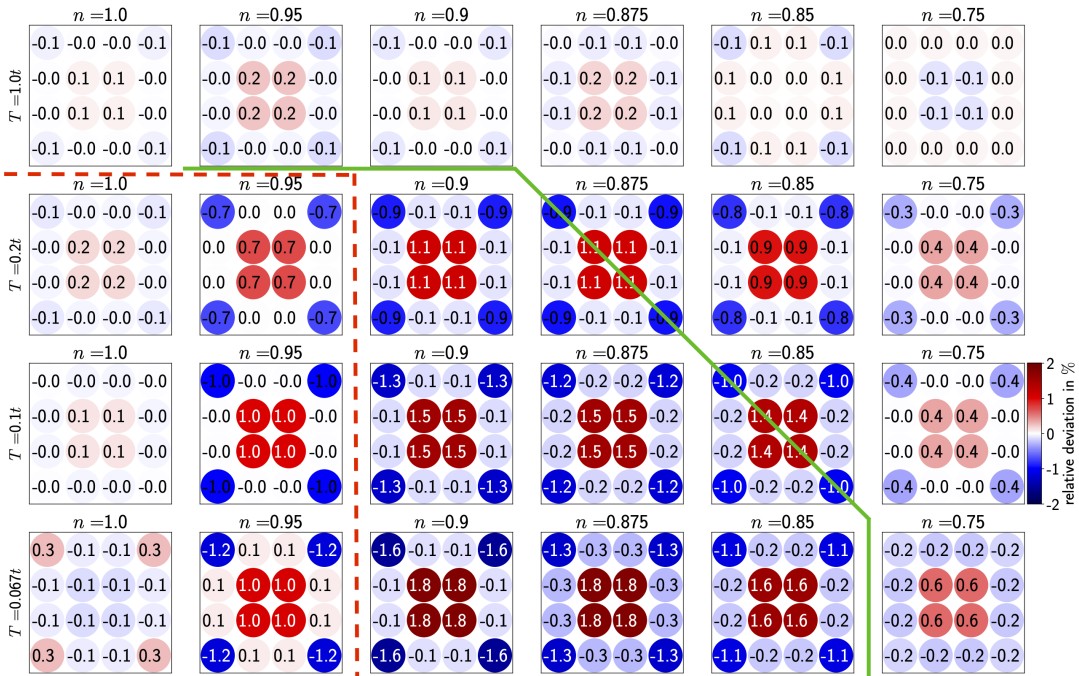

Figure 10: Relative deviations in per cent of the site-resolved particle densities $n_i = n_{\uparrow,i} + n_{\downarrow,i}$ from the total cluster filling for different fillings $n$ and temperatures $T$ ($U = 5t$, $t' = -0.25t$, $N_c = 4 \times 4$), rounded to 0.1%. Below the red dashed line the Fermi surface is renormalized, the green solid line marks the onset temperature of the pseudogap $T^*$, both determined from the lattice quantities in Fig. 9.

anti-nodal spectral weight from Matsubara data, as in Fig. 4 b). When extrapolating the pseudogap opening linearly to $T \rightarrow 0$ (grey, dashed), it coincides with the extrapolated trivial Lifshitz transition at $T \approx 0$ for $n \approx 0.78$. Let us finally remark, that no pseudogap has been found for the electron-like Fermi surface case.

Closing this section we compare the results of our center-focused CDMFT calculations with results recently obtained by Wu, et al. [78] within the dynamical cluster approximation (DCA) [15]. In contrast to CDMFT, in DCA the Brillouin zone is patched with momentum patches of constant self-energy. This construction does not break translation invariance, however, exhibits discontinuities at the patch boundaries. Also our calculations have been performed for $U = 5t$ and $N_c = 16$, while the DCA data have been obtained at $U = 7t$ and $N_c = 8$. Nonetheless the comparison of our results extrapolated to $T \rightarrow 0$ to the DCA results (shown as bottom bar of in Fig. 9) shows, despite the different technique and interaction value used, overall a good qualitative agreement with respect to the doping value of the Lifshitz transition and the onset of the pseudogap. Furthermore we point out that, as already observed in Ref. [78] a pseudogap only occurs for a hole-like Fermi surface topology.

## 5.5 Tendencies towards charge modulations

As the $N_c = 4 \times 4$ real space cluster offers the possibility of charge (density) inhomogeneities with respect to non-equivalent cluster sites, as a final analysis, we investigate the model's tendencies towards charge modulation. Previous studies using fluctuation diagnostics methods have shown that in the two-dimensional Hubbard model the pseudogap is opened at a temperature scale $T^*$ due to short-ranged spin fluctuations [80–82]. This does not exclude, however, ordering tendencies of the charge sector when lowering the temperature $T < T^*$ into the

pseudogap regime. Particularly interesting in this respect is the possibility of a stripe-ordered ground state of the model (i.e., the simultaneous occurrence of spin and charge modulations, see [83, 84], and, for other techniques and a broader overview [11] and references therein). The exact way how such a transition or crossover from a partially gapped Fermi surface to the charge-modulated state occurs and its possible interplay with superconductivity is not settled, and is, therefore, a topic of great current research interest, see for instance [68, 85–87].

In Fig. 10 we report the relative deviations of the site-resolved particle densities $n_i = n_{\uparrow,i} + n_{\downarrow,i}$ from the total cluster filling, again for $U = 5t$ and $t' = -0.25t$, calculated with our $N_c = 4 \times 4$ cluster. At high temperatures $T = 1.0t$, i.e., well above $T^*$ (indicated by the green solid line), only small deviations appear, which we attribute to the statistical noise of our Monte Carlo procedure. Interestingly, for lower temperatures $T < T^*$, the deviations become significant.[6] Tracing the level of charge inhomogeneity at the (lowest) constant temperature $T = 0.067t$ upon doping reveals a maximum well inside the pseudogap regime: before the Fermi surface reconstruction to hole-like (and the concomitant opening of the pseudogap), the inhomogeneity is relatively small. As soon as the pseudogap has fully developed, however, the inhomogeneity acquires its maximum before decreasing again approaching the Fermi-liquid regime. Hence, the intensity of charge inhomogeneities, remarkably, follows the same shape as the one of the pseudogap regime.

Our findings hint towards charge modulations inside the pseudogap regime $T < T^*$, however, future studies have to clarify the impact of finite-size effects (by means of larger clusters), as well as the exact nature of the putative crossover/transition (by means of two-particle susceptibilities).

# 6  Conclusions

In this paper we systematically analyzed the single-particle properties of the normal state of the two-dimensional Hubbard model on a square lattice utilizing CDMFT with center-focused post-processing.

In the half-filled particle-hole symmetric case, we could trace the progressive reduction of the onset interaction of the Mott metal-insulator crossover with increasing cluster size $N_c$. Remarkably, for cluster sizes $N_c \geq 16$ we could not detect a thermodynamic transition anymore for the accessible temperatures. This calls for further investigations of lower temperatures and, possibly, even of the ground state properties of the model (see also recent considerations from cluster perturbation theory [88, 89]).

For the doping-driven metal-insulator crossover we analyzed in detail the origin of the suppression of the antinodal spectral weight and the Fermi surface reconstruction by carefully investigating the (real space) contributions to the (momentum-resolved) self-energy at $U = 5t$ and $N_c = 4 \times 4$. We determined the location of the pseudogap opening and the Lifshitz transition discriminating a momentum-differentiated renormalized metallic from a conventional (isotropic) Fermi-liquid regime. For $t' \neq 0$ and half filling we find a strongly renormalized metallic state with an electron-like Fermi surface, hinting towards the presence of a Luttinger surface. Upon hole doping we observe a change in topology to a hole-like Fermi surface with a pseudogap. We were able to attribute this change in topology to the change of sign of the non-local contributions to the real part of the self-energy. Larger dopings lead to a regime with renormalized Fermi-liquid properties, and, eventually, to a second Lifshitz transition with an electron-like topology of the Fermi surface. We observe that the difference in doping of the pseudogap opening to the one of the Lifshitz transition decreases by lowering the tempera-

---

[6]The standard deviation of the individual density measurement is $\sigma < 0.3\%$ of the absolute value for $T = 0.067t$ at all fillings. Details can be found in Appendix D.

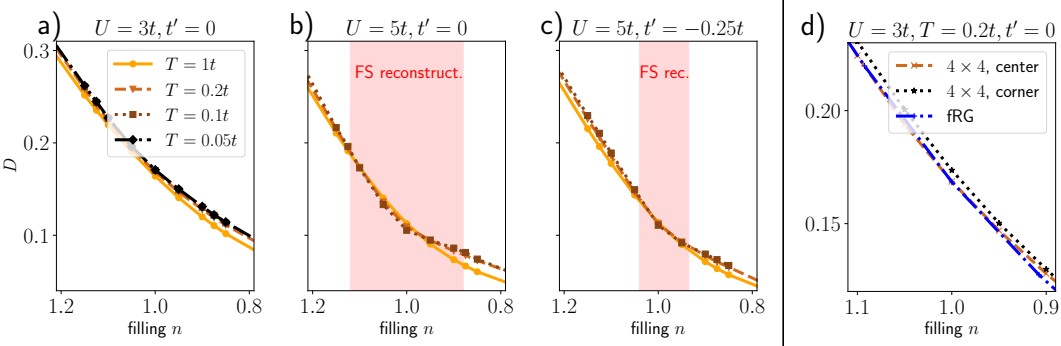

Figure 11: a)–c) Values for the double occupancy $D$ for various model parameters as a function of filling $n$ and temperature $T$, calculated at the center site of a $N_c = 4 \times 4$ cluster in CDMFT for a) $U = 3t$, $t' = 0$, b) $U = 5t$, $t' = 0$, and c) $U = 5t$, $t' = -0.25t$. The red shaded areas indicate regimes of a Fermi surface reconstruction (see text). d) Comparison of the double occupancy obtained from CDMFT with fRG for $U = 3t$, $t' = 0$, and $T = 0.2t$.

ture of the system. In contrast, no pseudogap has been found upon electron doping for our paramagnetically restricted calculations at $U = 5t$. Our results are in good qualitative agreement with previous studies utilizing the dynamical cluster approximation [78] at non-zero temperatures.

Eventually we found hints of emerging charge modulation via enhanced charge inhomogeneities in the pseudogap regime of the phase diagram. Whether a true transition to a charge ordered state is occurring has to be clarified by future studies based on the calculation of two-particle observables, going beyond the scope of this work.

# Acknowledgments

We thank Nils Wentzell, André-Marie Tremblay, Alessandro Toschi, Fedor Šimkovic IV, Daniel Rohe, Michel Ferrero, Pierre-Olivier Downey, Marcello Civelli, Mário Malcolms de Oliveira, Olivier Parcollet, Pietro Bonetti, Demetrio Vilardi, Nicklas Enenkel, Markus Garst, and Patrick Tscheppe for insightful discussions.

**Funding information** We thank the computing service facility of the MPI-FKF for their support. Michael Meixner gratefully acknowledges financial support by the Konrad-Adenauer-Stiftung e.V.

# A Double occupancy and comparison to functional renormalization group data

In this Appendix we show data for the double occupancy $D = \langle n_\uparrow n_\downarrow \rangle$ calculated for the center site of a $N_c = 4 \times 4$ cluster for various parameters, and eventually compare to results from the functional renormalization group (fRG) [90, 91].

In Fig. 11 a)–c) we show $D$ for different interaction strengths $U$ and values of $t'$ as a function of the filling $n$ and for several temperatures $T$. For all cases we can observe that, at fixed $T$, the double occupancy decreases monotonically upon hole doping. However, if the filling $n$

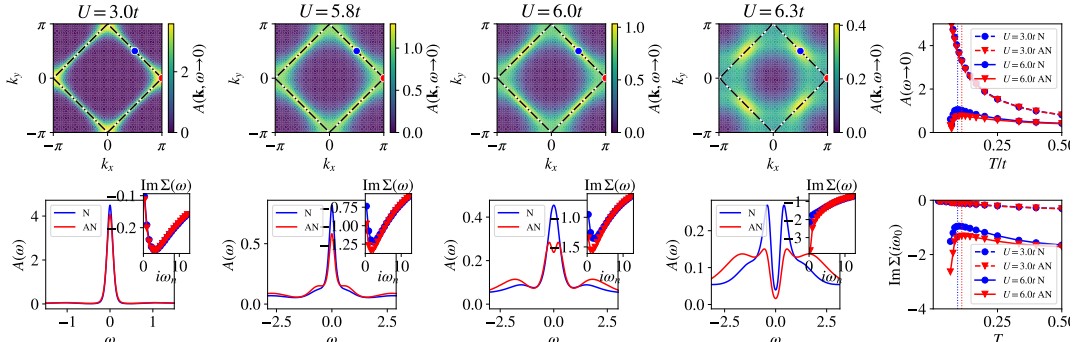

Figure 12: Results for the $2 \times 2$ cluster at half filling extracted from the periodized self-energy. a)–d) Spectral weight at the Fermi energy throughout the Brillouin zone for different values of the interaction $U$ at fixed temperature $T = 0.1t$. e) Spectral weight at the Fermi energy at the node and anti-node as a function of temperature for two different values of $U$. f)–i) Analytical continuation of the spectral function at the node and anti-node for the same parameters as in a)–d) in the row above. The insets show the imaginary part of the periodized self-energy for the first few Matsubara frequencies. j) Imaginary part of the periodized self-energy at the first Matsubara frequency as a function of temperature for two different values of $U$.

is kept constant, the weak interaction regime [$U = 3t$, panel a)] has to be discriminated from the intermediate interaction regime [$U = 5t$, panels b)–c)]: whereas in the former cooling the system from $T = t$ always results in an initial increase of $D$ (the famous Pomeranchuk effect in liquid $^3$He [47, 92, 93]), in the latter there are filling intervals where this effect is absent. Interestingly, these intervals centered around half filling agree with the parts of the phase diagram where the non-interacting Fermi surface is reconstructed in the periodized data (red shaded areas for $T = 0.1t$, see Sections 4.2 and 5.2).

Eventually, in panel d) of Fig. 11, we compare our results obtained from CDMFT at $U = 3t$, $t' = 0$, and $T = 0.2t$ to the ones obtained by the fRG.[7] Remarkably, the values for the double occupancy at the center site are in perfect agreement with the ones of fRG whereas the one obtained for the corner site are significantly larger. Given the fact that the fRG is fairly accurate at weak interaction strengths, this makes another case in favor for the center-focused post-processing of CDMFT [38].

# B  Additional analysis of small clusters

## B.1  Plaquette results for the Mott transition at half filling

In this Appendix we present data obtained from the periodization of a $N_c = 2 \times 2$ plaquette cluster for the half-filled Hubbard model on a square lattice with $t' = 0$. As in Sec. 3 we used the self-energy periodization in order to be able to compare to the results on larger cluster presented there. The top row of Fig. 12 shows the spectral intensity at the Fermi level $A(\mathbf{k}, \omega \to 0)$ for several values of the interaction at $T = 0.1t$. At low interaction strength $U = 3t$ we have a fully developed, connected Fermi surface. For both, the antinode and the node[8] clear quasipar-

---

[7]Here we employ the one-loop fRG with Katanin-substitution [94] and a frequency cutoff, see [42, 47, 95] for the details of the algorithmic implementation.

[8]In the case of this fully particle-hole symmetric case, their locations in momentum space are fixed at $\mathbf{k} = (\pi, 0)$ and $\mathbf{k} = (\pi/2, \pi/2)$, respectively.

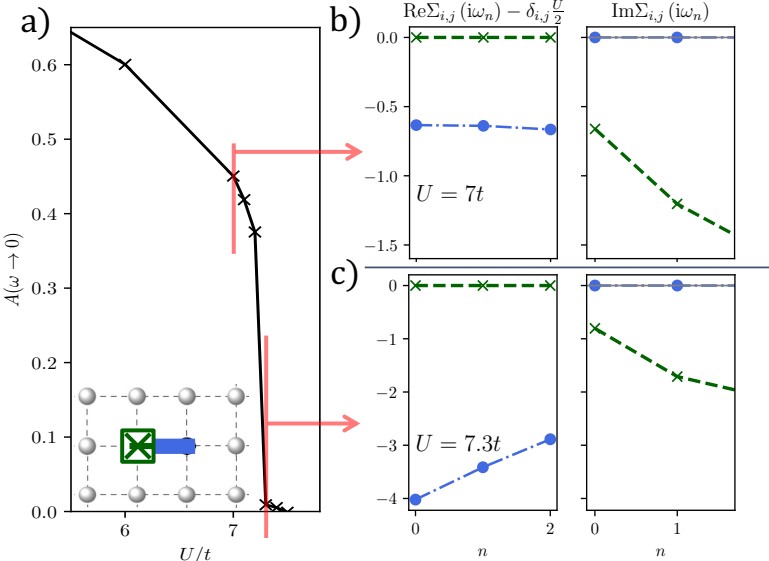

Figure 13: a) Spectral weight as a function of interaction U at $T = 0.05t$ for the $2 \times 1$ CDMFT at half filling. b) gives the self-energy short before the MIT at $U = 7t$ and c) short after the MIT at $U = 7.3t$. The green lines with "x" markers give the on-site self-energy while the blue lines with circular markers give the nn self-energy.

ticle peaks are visible in the analytically continued spectra (bottom row). As already pointed out in Sec. 3, the region of the momentum-differentiated breakdown of the metallic regime ("pseudogap") is rather narrow for the plaquette cluster ($5.9t \leq U \leq 6.1t$) compared to the one of the $4 \times 4$ cluster. The (one-particle) spectral key feature of the pseudogap, i.e., the increase of the nodal spectral weight and a decrease of the antinodal when cooling the system (rightmost sub-figure in the top row), could hardly be observed. The almost instantaneous destruction of the quasiparticle peaks at node and antinode in the spectral function on the real axis further corroborates these findings.

## B.2 Non-local correlations in the $2 \times 1$ CDMFT (dimer)

The smallest cluster incorporating non-local correlations within CDMFT is the $2 \times 1$ dimer. As shown in Fig. 13 a) for $T = 0.05t$ and at half-filling, the dimer exhibits a sharp drop of the spectral function corresponding to a MIT at $U_{c2} \approx 7.25t$. The transition as a function of $T$ and $U$ has been mapped out in the phase diagram in Fig. 1. The self-energies on the metallic side of the MIT [shown in Fig. 13 b)] exhibit a renormalized non-divergent behaviour while the slope is zero or negative. In the insulating case, just after the MIT at $U = 7.3t$ [Fig. 13 c)], the on-site self-energy (green dashed line) exhibits a metallic behaviour still, however, the real part of the next-neighbour self-energy (blue dot-dashed line) exhibits an inverted slope and is of the larges magnitude. This illustrates the importance of non-local correlations to the MIT in the two-dimensional lattice and is in contrast to the single-site DMFT, where the imaginary part of the on-site self-energy diverges with the MIT, but at much larger interaction values. The MIT in DMFT is found at $U = 9.7t$ for this temperature.

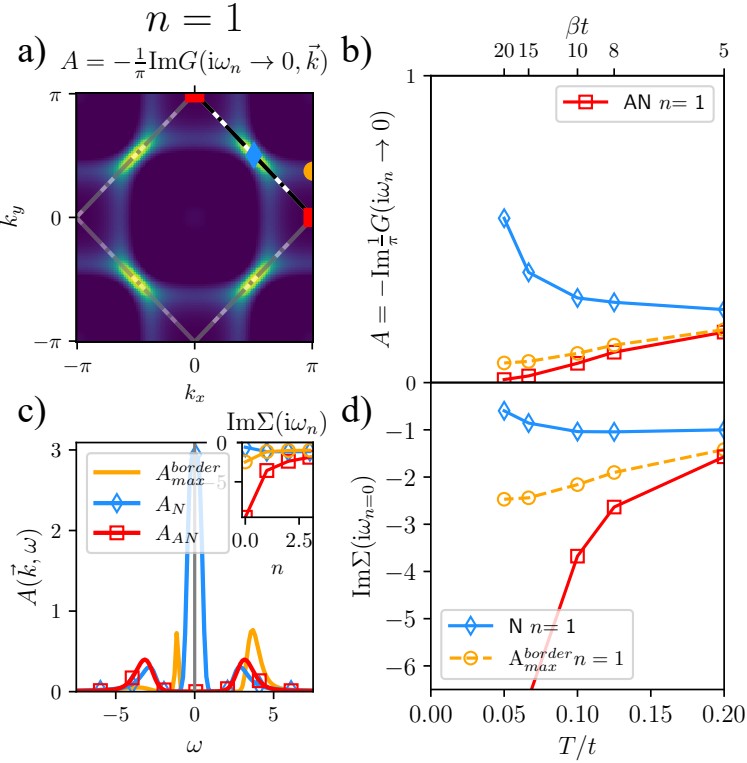

Figure 14: a) Fermi surfaces and spectral weights (color coded), extrapolated from the Matsubara axis for $U = 5t$, $t' = 0t$, and $T = 0.05t$ obtained from the $N_c = 4 \times 4$ cluster and center-focused periodization, for the half-filled case. White solid lines indicate the non-interacting Fermi surfaces, black dotted-dashed lines the respective interacting ones. b) The spectral weight at antinode and node as a function of the temperature and filling. c) Analytically continued [44] spectral function for the antinode (red), node (blue) and the maximum of the spectral weight at the edge of the Brillouin zone (orange). The insets show the imaginary part of the self-energies as a function of the Matsubara index for the antinode and node. d) Imaginary part of the self-energy at the first Matsubara frequency as a function of temperature. Red squares denote the antinode, blue diamonds the node throughout the figure.

## C  On the pseudogap in the particle hole symmetric case at $U = 5t$

The temperature dependent analysis of single-particle quantities in Sec. 4.2 is based on the definition of the antinode as the intersection of the solution to the quasi-particle equation with the border of the Brillouin zone, see Sec. 2.3. The spectral weight throughout the Brillouin zone in the half-filled case given in Fig. 4 shows a folding along the antiferromagnetic zone boundary, resulting in a maximum of the spectral weight along the edge of the Brillouin zone $A_{\max}^{\text{border}}$ not being at the previously defined antinode. Fig. 14 presents a comparison between the point $A_{\max}^{\text{border}}$ (orange), the antinode (blue) and the node (red). From b) we can infer a pseudogap regime due to a reduced spectral weight when reducing the temperature for both, the antinode and $A_{\max}^{\text{border}}$. The self-energy in c) is clearly divergent and the spectral function from analytic continuation [44] displays no quasi-particle peak, a strong upper Hubbard band and a weak lower Hubbard band. A strong peak at the upper edge of the lower Hubbard band becomes visible, hinting towards a strong doublon-holon mechanism [96]. We clearly see the non Fermi-liquid like behaviour from the non-linear behaviour of the self-energy in d).

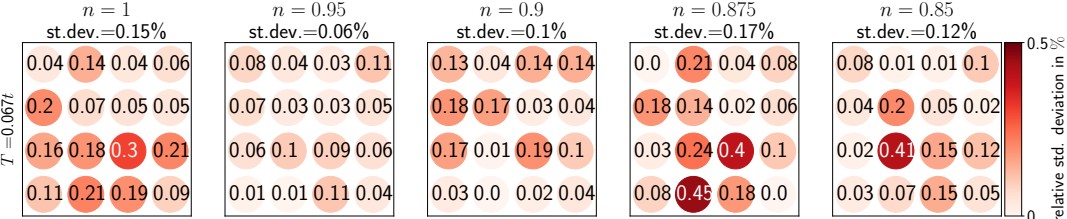

Figure 15: Relative deviation of the "raw" site-resolved densities from the symmetrized cluster result for different dopings at $T = 0.067t$. We also indicate the standard deviation of the entire cluster.

## D    Statistical significance of the charge modulation analysis

This Appendix addresses the statistical significance of the data presented in Section 5.5, where the modulation of the charge density on the cluster was discussed. We focus on $T = 0.067t$. There, inside the pseudogap regime between $0.95 > n > 0.75$, deviations of the site-resolved density to the cluster filling $> 1\%$ were observed. Fig. 15 presents an analysis of the noise on the density measurement for these fillings for $T = 0.067t$. Indicated in red color coding are the deviations of the raw results of the site-resolved densities from the results symmetrized over all respective equivalent cluster sites, which gives a proxy of the statistical sampling error. We find that these deviations are in the order of $0.2 - 0.45\%$, giving a standard deviation of $\sigma_{\text{st}} < 0.2\%$ for the entire cluster. Hence, statistic variations are considerably lower than the relative deviations presented in Section 5.5, underlining their statistical significance.

## E    Cluster sketches

This Appendix provides a description of the cluster geometries analyzed in this paper, which are sketched in Fig. 15. a)–c) show the $1 \times 1$, $2 \times 1$ and $2 \times 2$ (C)DMFT, where all cluster sites are equivalent. Fig. 15 d) provides the sketch of the $4 \times 4$ CDMFT cluster. Here, not all cluster sites are equivalent anymore. The sketch discriminates the sites of the outer rim of the cluster, consisting of edge- and corner sites, and the center sites, which are separated by a blue dashed line.

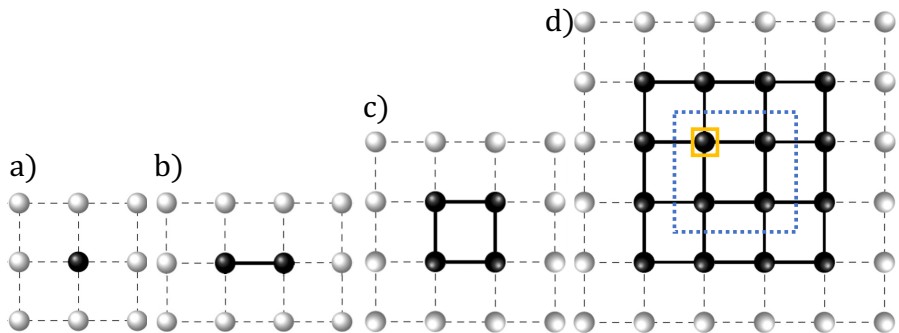

Figure 16: Sketches of the different cluster geometries discussed throughout this paper: a) the $1 \times 1$ CDMFT (i.e DMFT), b) the $2 \times 1$ CDMFT (i.e dimer), c) the $2 \times 2$ CDMFT (i.e plaquette), d) the $4 \times 4$ CDMFT, where the blue dashed discriminates between the outer rim and the center sites and the yellow square marks a "center-site".

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
