# Peer review of "Mott transition and pseudogap of the square-lattice Hubbard model: results from center-focused cellular dynamical mean-field theory"

_SciPost Physics, doi:SciPost Phys. 16, 059 (2024)_

## Round 1 · Referee Report · Pierre-Olivier Downey (Referee 1) · 2023-11-23

Strengths

1- A thorough study of the square Hubbard model is conducted, comparing a wide range of clusters. The article presents the results clearly, establishing it as a valuable reference for those investigating the MIT.

2- A systematic examination of the doping phase diagram for a slightly frustrated $4\times4$ cluster is undertaken. This enables the investigation of density variations between central and border sites.

3- The article illustrates the transformation of the Fermi surface from e-like to p-like and explores its evolution with temperature.

4- The authors demonstrate the probable existence of charge density waves below $T^*$ with CDMFT.

Weaknesses

1- While the frustrated square lattice is explored with $t'=-0.25$, the authors do not specify the $U$ at which the pseudogap emerges at half filling, or whether $U>U_{MIT}$. Consequently, readers can only speculate about their wherabouts.

2- With a higher second neighbor hopping term, as demonstrated in reference [24], frustration may impede the disappearance of the MIT in larger clusters. It remains unclear if a first-order MIT can be observed with $t'=-0.25$ at the temperatures studied.

3- Although the authors show that border sites converge slower with cluster size than center ones, they use the increased difference between center sites and border one to conclude that there might be a charge density wave.

Report

This paper presents a comprehensive analysis of the phase diagram at half filling for the square, unfrustrated lattice. They strongly suggest that no first-order transition exist at the thermodynamical cluster-size limit, instead showing signatures of a pseudogap at half filling, or at least, the folding of the Fermi surface. Upon doping the system, the paper demonstrate a clear pseudogap solution and a Lifshitz transition. From Fig. 2, the paper provides clear evidence that the central site converges faster then corner sites in CDMFT. Fig. 1 answers to a long-standing question : the behavior of the Mott transition in the large $N$ limit.

With $t'=-0.25$, a pseudogap is also shown, as well as a Lifshitz transition. They find a hole-doped pseudogap and present how the pseudogap evolve with respect to temperature and doping. They also show that there is an electron-like Fermi surface at low doping, without a pseudogap phase, which agrees very well with results from Ref. [75] that showed that no pseudogap forms while the Fermi surface is e-like.

They close their article by showing possible hints of charge density wave far below $T^*$, suggesting a charge density wave ground state. This discovery does, at the same time, open the possibility of using smaller-than-before[83] clusters ($4\times 4$) to study this strange phase. This shows that the study of charge density wave near the Mott transition may be possible.

It is worth noting that the faster convergence of central sites may be the cause for what appears to be the charge order, I think this suggest that further studies are necessary. Finally, this article answers every general acceptance criteria, plus, this work satisties both criteria : "Present a breakthrough on a previously-identified and long-standing research stumbling block" and "Open a new pathway in an existing or a new research direction, with clear potential for multipronged follow-up work". I strongly recommend its publication in Scipost

Requested changes

My first requested changes are aethetic changes, thus not an obligation, although I believe they would give further value to the article.

1- Page 3 line 11, a typo was probably introduced. I believe it should be "serves" instead of "servers".

2- Maybe these are is just a details, but maybe the $T^{4x4}_{Néel}$ and $T^{6x6}_{Néel}$ should be written with $\times$ instead of $x$.

3- Maybe the color of $T^{6x6}_{Neel}$ should be the same as the $6\times6$ Widom line.

The next requested changes are more important

4- In page 9, the authors discuss the pseudogap at half filling, but Fig. 4 a) seems to suggest that there is a strong folding of the Fermi surface. In my opinion, such behavior is associated to an AFM phase and may not account for a pseudogap phase, unless the "Fermi surface" of those pockets near the border of the Brillouin zone are shown to have no coherent quasi-particles. Can the author confirm this and mention it in the text ? (I don't think this needs a new figure, but I believe this needs to be mentionned)

5- Perhaps I missed it in the text, but for Fig 4 a) and b), how do the authors make the extrapolation of the imaginary part of the Green's function near $\omega=0$? I guess it is a polynomial fit, but what is the order?

6- Both Fig. 7 and 9 are consistent, relating to the inexitence of a pseudogap at half filling. What I find quite surprising is that this does not match what is usually seen in the litterature. Particularily, - Ref.[75] shows that the pseudogap on the electron doped side should survive up to 4% of doping at $U=7.0$ and $t'=-0.25$ (Figure 2, $t'=0.25$, using pte-hole transformation). - Ref. PRB 71, 134527 (2005) shows that the two and one band Hubbard model (Fig. 4 and 14) have an electron doped pseudogap - Ref. [24] shows the existence of $T^*$ on the triangular lattice at half filling. - Ref. PRB 89, 245130 (2014) shows that the anisotropic triangular lattice does have a pseudogap at half filling. The fact that you don't find it is, in my opinion, probably because $U$ is too small to see it. This may seem unrelevant to the initial goal of the article, but I believe that a small comment on that is necessary (probably at the end of Sec. 5.4). Many believe that the electron doped pseudogap only exists in the low-interaction regime, and I wouldn't want people to misinterprete your figures, concluding that there might be no electron-doped pseudogap in the strong coupling regime.

7- On Fig. 10, the authors should mention that the values are given in "%" since I was confused during my first reading.

8- Maybe I missed the explaination in the text, but why can you conclude that there is charge density wave while it is shown earlier that central sites converge faster to the infinite-size cluster MIT solution? Pseudogap solutions are usually considered as having increased effects from correlations, this might be the reason why the differentiation is higher in the pseudogap regime than in the Fermi liquid regime in Fig. 10. I think that the authors give a great explaination, but should mention the above interpretation, or give a small explaination on why faster convergence on central sites does not impact this.

  • validity: high
  • significance: high
  • originality: high
  • clarity: top
  • formatting: perfect
  • grammar: perfect

Author:  Thomas Schäfer  on 2024-01-17  [id 4253]

(in reply to Report 1 by Pierre-Olivier Downey on 2023-11-23)
Category:
answer to question

We thank Mr. Downey very much for his thorough reading of the manuscript, the valuable comments which helped us to improve our manuscript, and for strongly recommending the publication of our article in SciPost Physics. We address his points in the PDF attached.

Attachment:

Referee1_cfCDMFT-2.pdf

---

## Round 1 · Referee Report · Anonymous (Referee 2) · 2023-11-29

Strengths

  1. Nonperturbative investigation of Mott phase diagram in doped regime
  2. Numerically exact CT-INT cluster-DMFT solver
  3. Detailed analysis of nodal-antinodal dichotomy

Weaknesses

  1. CDMFT break translational symmetry, therefore the authors used "center-focused " approach

Report

The authors investigate correlation effects in 2d-Hubbard model using Cluster Dynamical Mean Field (CDMFT) scheme in so-called "center-focused" approach. Numerically exact Continuous Tome Quantum Monte Carlo (CTQMC) in interaction expansion (CTINT) approach was used.
Very detailed and accurate calculations of phase diagram for undoped and doped Fermi-Hubbard lattice as function of particle-hole asymmetry related with t'/t were carried. I find results are very interesting and shed light on metal-insulator crossover in the doped Hubbard model and tendency toward charge-density waves.
I support the publication in SciPost Physics.
  • validity: good
  • significance: high
  • originality: high
  • clarity: top
  • formatting: excellent
  • grammar: excellent

Author:  Thomas Schäfer  on 2024-01-17  [id 4254]

(in reply to Report 2 on 2023-11-29)

We thank Reviewer 2 for her/his comments and reviewing the manuscript. Reviewer 1 does not request any changes and supports the publication in SciPost Physics.

---

## Round 1 · Referee Report · Anonymous (Referee 3) · 2023-12-19

Strengths

1- The article extensively study the Hubbard model for different large cluster sizes, interaction strength, doping and frustration.

2 - The comparison at half-filling with the phase diagram at different cluster sizes and method is extremely interesting and revealing the limitations of small cluster studies.

3- The work presents many results in a clear way.

Weaknesses

1- The methodology results to concise and it deserves additional explanation.

2- Periodization technique should be compared with other methods like the cumulant periodization approach and the result obtained compared with the coarse grained Fourier result within the cluster

Report

This work is very impressive in terms of simulations done for several different parametrical regimes and I congratulate the author for the quality of their job.

I appreciate the systematic comparison in figure 1 for several different sizes and methods. It would have been nice to see something similar for the doping study.

Requested changes

1- I would suggest to detail the number of CPU hours used in order to obtained the overall results.

2- In order to determined Tneel is not possible to compute a stager magnetization within the cluster as order parameter? Would it lead to the same critical temperature?

3- I would suggest a very recent work to the authors for this half-filled analysis: Local and nonlocal electronic correlations at the metal-insulator transition in the Hubbard model in two dimensions, Maria Chatzieleftheriou, Silke Biermann, Evgeny A Stepanov

4- The periodization procedure chosen should be compared with the exact result of the Coarse grained Fourier transform within the cluster. I also wonder in respect to the methodology, if the choice of the site, where the analysis is performed, is equivalent to other choices of site within the centrals one. Would be worth to create a local self-energy of just central sites? The Self-energy are presented always for the first Matsubara frequencies but what is their behavior before imposing the cut-off at high frequencies?

5- In Fig. 2 and then in the text there are a series of abbreviation not introduced in the extended form like nn nearest neighbor and so on.

6-what kind of procedure MaxEnt was used in order to obtained real frequencies data?

7- Fig.10 is somehow confusing. I would expect that by summing up all contributions of the deviation we should obtain 0, but this is not always the case. How is the occupation of the system computed? Couldn't be computed as an average over the occupation per-site?

8- The use of reference 32 as example of the study of SC phase should be amended given that the results presented problem of Ergodicity (Sémon, P., G. Sordi, and A-MS Tremblay. "Ergodicity of the hybridization-expansion Monte Carlo algorithm for broken-symmetry states." Physical Review B 89.16 (2014): 165113)

  • validity: high
  • significance: high
  • originality: top
  • clarity: high
  • formatting: excellent
  • grammar: excellent

Author:  Thomas Schäfer  on 2024-01-17  [id 4255]

(in reply to Report 3 on 2023-12-19)

We thank Reviewer 3 very much for her/his thorough reading and the valuable comments, allowing us to improve on our manuscript. We further thank Reviewer 3 for her/his appreciation of our work. We address her/his comments in the PDF attached.

Attachment:

Referee3_cfCDMFT.pdf

---

## Round 2 · Referee Report · Pierre-Olivier Downey (Referee 1) · 2024-1-17

Report

I believe the paper is ready to be published.

---

## Round 2 · Referee Report · Anonymous (Referee 3) · 2024-1-20

Report

The article is ready for publication

Requested changes

No additional changes request

---

## Round 2 · Referee Report · Anonymous (Referee 2) · 2024-1-25

Report

The Authors reply to all my comments.
I recommend this paper for publication.

---

## Round 2 · Author Response

We thank all three Reviewers for their thorough reading of our manuscript and for very useful comments. Given the positive remarks of the Reviewers and the addressing of all their comments and remarks in the new version of our manuscript and the replies, we feel that our paper is now ready for publication in SciPost Physics.

---

## Round 2 · List of Changes

On request of Referee 1:
-) Correction of a typo on page 3.
-) Style change in Figure 1.
-) A footnote on page 9 discussing pseudogap physics in the particle-hole symmetric, half-filled case and referring to the following amendment.
-) Amendment of a new Appendix (C), discussing the pseudogap in the particle hole symmetric case at U=5t, including a new figure, Fig. 14, on page 22f.
-) A footnote on page 16, discussing the absence of a pseudogap for the half-filled and electron doped case. Also, citations were added.
-) Amendment of the caption of Fig. 10.

On the request of Referee 3:
-) A footnote on page 4 specifying some numeric cost.
-) A short comment, mentioning the suggested literature by [Chatzieleftheriou et al.] at the end of section 3.1 (page 7).
-) Rementioning the definition of 2nn as second-nearest neighbor at the beginning of section 5, page 12, for readability.
-) A more detailed description of the MaxEnt-procedure used in the calculations, including a link to the used package on page 5.
-) An amendment of the caption of Fig. 10 to detail the rounding of the given numbers to 0.1%.
-) Amendment of the reference given by the Referee.

---

## Editorial Decision

published